# Relationship between Self-Concept, Emotional Intelligence and Problem-Solving Skills on Secondary School Students' Attitude towards Solving Algebraic Problems

Abdul Halim Abdullah [1,*], Elizabeth Julius [2], Nornazira Suhairom [1], Marlina Ali [1], Corrienna Abdul Talib [1], Zakiah Mohamad Ashari [1], Umar Haiyat Abdul Kohar [3] and Sharifah Nurarfah S. Abd Rahman [1]

[1] School of Education, Faculty of Social Sciences & Humanities, Universiti Teknologi Malaysia, Johor Bahru 81310, Malaysia
[2] Department of Science Education, Faculty of Education, Kebbi State University of Science & Technology, Aliero 863104, Nigeria
[3] Faculty of Management, Universiti Teknologi Malaysia, Johor Bahru 81310, Malaysia
* Correspondence: p-halim@utm.my

**Abstract:** Mathematics is required from primary (basic) through junior secondary and senior secondary levels of education in Nigeria. The position of mathematics within the curriculum is reflective of the significance of the subject to the expansion of scientific knowledge and technological capability. However, student performance in mathematics, particularly algebraic aspects, is a challenge for Nigerian secondary school students. The purpose of this study was to investigate the relationship between self-concept (SC), emotional intelligence (EI), and problem-solving skills (PSS) on students' attitudes towards solving algebraic problems (ATSAP). The study aimed at developing a model based on students' SC, EI, PSS, and ATSAP. A total of 377 students were proportionately and randomly selected to collect the quantitative data. Two instruments: (i) a questionnaire measuring SC, EI, and ATSAP, and (ii) a test measuring PSS developed by the researcher, were used in this study. The data was analyzed by using structural equation modelling (SEM) and partial least square (SEM-PLS3). The major findings of the study revealed that secondary school students' PSS could be improved with due consideration of their SC and EI. However, the three hypotheses tested indicated that there was a significant but negative relationship between SC and the students' ATSAP (t > ±1.96, $\alpha$ = 0.05) and also there was a significant positive relationship between PSS and their ATSAP; however, the relationship between EI and their ATSAP was not significant (t < ±1.96, $\alpha$ = 0.05). Therefore, a model was developed based on the study's findings. This model has a practical implication for the federal and state governments, curriculum planners, students, teachers, and parents.

**Keywords:** structural equation modelling; self-concept; emotional intelligence; problem-solving skills; attitude towards solving algebraic problems

## 1. Introduction

In the Nigerian education system, mathematics is a mandatory subject to be offered from primary (basic) through the junior secondary and senior secondary levels of learning. The craving for a high level of performance in the subject placed a lot of pressure on students, teachers, parents and, specifically, the school and, in general, the education system itself. In Sokoto state, like most states in Nigeria, mathematics is a compulsory subject from primary through to secondary school level. The importance of mathematics in the school curriculum cannot be compared with any other subject in the field of education. This is because mathematics has played a big role in developing human thought, it also brought about systematic reasoning processes used in problem solving and analysis [1].

The key anticipation of mathematics as a subject is the development of problem-solving skills, as stipulated by [1,2]. Mathematics is an instrument that can be applied to

train students to be able to solve problems and also to build their thinking abilities that lead to further solving of non-mathematical problems. It is imperative that learning to solve problems be fundamental to learning mathematics, as problems are part of everyday life. Problem solving, according to [3], is a cognitive process directed at achieving a goal when no solution method is obvious to the problem solver. Their definition consisted of four parts: Firstly, problem solving is cognitive, i.e., it occurs within the cognitive system of the problem solver. Secondly, problem solving is a process, i.e., it involves applying a cognitive process to cognitive presentation in the problem solvers' cognitive system. Thirdly, it is directed, i.e., it is guided by the problem solver, and lastly, it is personal, i.e., it depends on the knowledge and skills of the problem solver.

Therefore, the ultimate aim of teaching and learning mathematics at any level is to solve problems [4]. Ref. [4] described it as the backbone of science and technology and an inevitable tool for human survival in everyday life. Secondary school mathematics has many core areas of study, such as algebra, mensuration, statistics, numbers and numeration, geometry, and trigonometry. All the aforementioned aspects of mathematics involve the application of problem-solving skills.

In algebra, letters stand in for numbers. According to [5], algebra is a branch of mathematics that makes use of both alphabets and arithmetic. Having letters and numbers together is already confusing enough, but when the letters' values shift or one letter is substituted for another at intervals, it becomes even more so for students. What is done to one side of an algebraic equation with a number on one side of the scale is also done to the opposite side of the scale [6]. The numbers are fixed; they span the gamut from real to complex numbers to matrix to vectors.

This has led to many students failing to achieve a basic algebraic literacy, as such, it turns out to be a barrier to their entry into careers in sciences, engineering, technology and business [6–10]. Therefore, all mathematical processes of problem solving and all problems are resolved into algebraic expressions and equations for possible solutions. Middle-school teachers frequently select tasks that can be used to improve algebraic reasoning. It is critical that teachers learn to recognize the algebra that occurs naturally in focusing tasks on other strands and transform those occurrences into meaningful learning opportunities. Looking at EI as one of the social skills, students will be better prepared for the mathematical challenges that lie ahead if teachers pay attention to the algebraic opportunities that already exist in those tasks and modify those tasks so that they foster algebraic thinking among students. This can go a long way toward ensuring that students are successful in their future mathematical endeavors [11]. This implies that, it is only those students who understand, master, and retain knowledge and skills in algebra that are likely to apply them successfully in their mathematics field and in real life. Research has revealed that several factors play a vital role in influencing students' understanding of algebraic expressions and performance in mathematical problem solving, such as students' self-concept, self-efficacy, extrinsic motivation, experience in school, and attitudes [12–14].

Research studies have indicated that students' attitudes towards mathematics, and algebra in particular, are very much correlated to their attitude towards problem solving in general [15–17]. Ref. [18] suggested that negative attitudes in students needed to be overcome in order to prevent the persistent occurrence of poor algebraic problem-solving skills in the future. Algebra is a challenging area of mathematics education. However, students cannot avoid learning algebra. This is due to the fact that algebra is often a prerequisite for entry into fields of study and careers that require higher sophistication. In addition, the goal of algebra is to give students a chance to hone their deductive reasoning skills. This goes against the grain of what is taught in schools these days about mathematics. The majority of algebra instruction in schools nowadays is devoted to an emphasis on procedural-based topics, unrelated to real-world problems. This goes against the purpose of learning math in general and algebra in particular, which is to equip pupils to solve issues in an increasingly complicated environment [19]. It is imperative to teach students how to overcome the mathematics representation phase since many difficulties they face are not

connected to algebraic problem solving. Students' interest in mathematics and their beliefs in the usefulness of mathematical knowledge in their future career or in their everyday life are determined in a fundamental way by their problem-solving behavior [20–22]. Therefore, students' attitudes towards solving algebraic problems is largely affected by their beliefs on how important mathematics is to them and by the level of the students' confidence in solving problems in mathematics.

Algebraic problem solving in mathematics as a subject is of vital importance [23]. Students equipped with problem-solving skills would increase their self-confidence, which directly affects the way they see and evaluate themselves in learning mathematics. The evaluation usually leads to them to have a positive or negative self-concept. Self-concept is a domain-specific self-evaluation [24], a thorough representation of an individual's beliefs, which has influenced many disciplines, including social psychology, personality, education, child development, mental and physical health, social service, organization, industry, and sport [25]. Stedman's medical dictionary defines self-concept as an individual's idea of self, including an appraisal of one's standing based on society or personal norms. Ref. [26] defined self-concept as the sum of an individual's beliefs, attitudes, and opinions about him/herself.

Ref. [26] characterized the self as perceptual, conceptual, and attitudinal. The perceptual component is a person's image of his body's appearance and the impression he makes on others. The conceptual component is a person's notion of his particular qualities, abilities, background, origin, and future. This is called psychological self-concept and includes honesty, self-confidence, independence, courage, and their opposites. The attitudinal component refers to a person's feelings about himself, his present status, and future prospects. Early adolescence is a pivotal phase in children's self-concept development [27], as it profoundly modifies the sense of self. The child's self-concept develops into an increasingly sophisticated and integrated self as a result of the environment and social interaction [26,28]. Self-concept is the basis for all motivational behavior in an individual. Generally, the construct of self-concept has been utilized in several studies as a measure of the cognitive (intellect and problem-solving ability) and affective (emotions, such as motivation, interest, anxiety, belief, and attitude) parts of an individual [29–33]. In another study, self-concept was found to be essential in building students' emotions [34].

Emotions in an individual are considered fundamental because they underlie every expression of evaluation in one way or another [35]. Emotions develop into students having strong or weak emotional intelligence (EI). EI, as one of the variables in this research, is a new construct in psychology, academic performance, social skills, career, marriage, and personal life. It has to do with the ability to accurately perceive emotions, to access and generate emotions to assist thinking, to understand emotions and emotional knowledge, and to regulate emotions in others to improve emotional and intellectual growth in an individual [36]. It is the set of abilities that account for how people's emotional reports vary in their accuracy and how the more accurate understanding of emotions leads to better problem-solving skills in an individual's life [37].

EI can be considered as a factor that can be learned and taught as an indicator that is capable of preserving and improving problem solving skills [38]. These demand teachers to create a classroom environment that integrates EI in planning mathematics instruction either through the students' activities or exercises, so as to improve the students' attitude towards problem solving in algebra [14]. To ensure that students succeed in introductory mathematics courses, it is critical to first understand how they relate algebraic operations to basic number properties. Students gain an understanding of algebraic operations, according to researchers, by making connections to basic number properties [39]. The findings of previous studies showed that students who have high emotional control can obtain high achievement in mathematics and science. This is because a student who has adequate management and control of emotions can excel in mathematics and science, which require high cognition load [38]. Similarly, ref. [40] found that students who followed the lessons by integrating EI showed significantly higher mathematics scores. Therefore, students who

had a high EI usually had a better recognition of their own feelings and emotions and those of others.

A study on the influence of EI on SC conducted by [34] found that EI was essential in building SC. Feelings and expressions of emotions appropriately exerted a great influence in having a positive SC. Considering the fact that problem-solving ability in algebra plays an important role in students' performance in mathematics, it is necessary to find out which factors could improve their problem-solving ability in algebra and how these factors relate to their attitudes towards problem-solving in algebra. The present study focused on four variables; self-concept (SC), emotional intelligence (EI), and problem-solving skills (PSS) on attitude towards solving algebraic problems (ATSAP). This study specifically sought:

i.    To determine which factor has an influence on students' attitude towards solving algebraic problems;

    a.    To determine the influence of self-concept (SC) on attitude towards solving algebraic problems (ATSAP) amongst secondary school students in Sokoto state.

    b.    To determine the influence of emotional intelligence (EI) on attitudes towards solving algebraic problems (ATSAP) amongst secondary school students in Sokoto state.

    c.    To determine the influence of problem-solving skills (PSS) on attitudes towards solving algebraic problems (ATSAP) amongst secondary school students in Sokoto state.

ii.    To examine the relationship between self-concept (SC), emotional intelligence (EI), and problem-solving skills (PSS) amongst secondary school students in Sokoto state.

iii.    To develop a model based on students' attitude towards solving algebraic problems (ATSAP) in relation to their self-concept (SC), emotional intelligence (EI), and problem-solving skills (PSS).

Two research questions were formulated based on these objectives:

i.    RQ1: Are there any significant influences of self-concept (SC), emotional intelligence (EI), and problem-solving skills (PSS) on attitudes towards solving algebraic problems (ATSAP)?

ii.    RQ2: Are there any significant relationship between self-concept (SC), emotional intelligence (EI), and problem-solving skills (PSS)?

From these research questions, six hypotheses were developed:

i.    Null Hypothesis 1: There is no significant influence that exists between self-concept (SC) and attitude towards solving algebraic problems (ATSAP) among secondary school students in the state of Sokoto.

ii.    Null Hypothesis 2: There is no significant influence that exists between emotional intelligence (EI) and attitude towards solving algebraic problems (ATSAP) among secondary school students in the state of Sokoto.

iii.    Null Hypothesis 3: There is no significant influence that exists between problem-solving skills (PSS) and attitude towards solving algebraic problems (ATSAP) among secondary school students in the state of Sokoto.

iv.    Null Hypothesis 4: There is no significant relationship that exists between self-concept (SC) and emotional intelligence (EI) among secondary school students in the state of Sokoto.

v.    Null Hypothesis 5: There is no significant relationship that exists between emotional intelligence (EI) and problem-solving skills (PSS) among secondary school students in the state of Sokoto.

vi.    Null Hypothesis 6: There is no significant relationship that exists between self-concept (SC) and problem-solving skills (PSS) among secondary school students in the state of Sokoto.

*Inconsistency in Findings*

Quite a lot of research has been done in relation to the variables under study. Scholars have studied the SC, EI, and PSS of students in mathematics in relation to their academic achievement. All of the findings in the trend of the previous research indicated inconsistent results. For instance, in the study carried out by [41] on the EI of students and their academic performance, the results indicated that there was no significant difference in all the dimensions of EI and their academic performance. Ref. [42] also found, in their study, that EI did not predict the academic achievement of students. On the contrary, in a study conducted by [43,44], a significant relationship between EI and academic achievement was shown. Ref. [45] found a significant low-positive relationship between the EI of students and their academic performance. From the aforementioned, it is evident that there is inconsistency in the findings of previous studies, therefore, making the present study necessary to be carried out, more so the present study will be in a different environment entirely with different subjects from a different culture and background, as suggested by [46]. Similarly, research on SC has also been done previously, where the study by [47] showed that SC was a strong predictor of academic achievement and that there was a relationship between SC and students' academic achievement. This result was in agreement with the findings of [33,48–55]. On the contrary, refs. [56,57] found that SC did not predict the students' academic performance.

However, although the study of attitudes towards mathematics has been developed over a long period of time, the study of attitudes towards solving algebraic problems has a short history in mathematics education and is a dearth in the literature. This study will attempt to investigate the role of SC, EI, and PSS as specific variables in order to identify the predictor of positive ATSAP. The purpose of this study, therefore, is to empirically examine the relationship between SC, EI, and PSS in relation to students' ATSAP, which has not been explored in detail. Most studies on SC and EI in the Nigerian context have largely examined these variables in relation to students' general academic performance and not to specific subject areas, whereby the linking process of both variables was largely neglected, especially the aspect of students' ATSAP, which was under-explored and unclear.

## 2. Conceptual Framework

The conceptual framework is a description of the main independent and dependent variables in a study and the relationship between them. Independent variables are conditions or characteristics that are manipulated to ascertain the relationship and observed phenomenon, while the dependent variables are conditions that appear to change as the independent variables are introduced or removed. The conceptual framework of this study is based on the theories, from which four constructs and eleven sub-constructs have been identified as proposed in Figure 1 (acted as a frame of reference to show the relationship between SC, EI, PSS, and ATSAP). As shown in Figure 1, all constructs were positively related. EI is vital to building SC, according to much research [19,34,44,58,59]. The focus of the study is on how emotions affect a person's mathematical behavior, as suggested by [60–62]. The study demonstrates how emotions can have varying influences on how a person's intrinsic ability (SC) develops. It is believed that performance can be improved by influencing emotions, cognitions, and behaviors.

Studies on the relationship between EI and PSS indicated that EI could manage and control emotions to a stable state of solving problems [20,38,63–66]. EI is a construct about an individual's skills that can help one to better understand how to make a meaningful perception and think logically by using emotions. Students with high EI were believed to possess a better capacity to perceive and reason about emotions, which facilitated a greater positive effect [67,68]. Similarly, previous studies indicated that SC is a driving force in PSS [51,69–72]. Previous research studies primarily focused on the relationship between the variables under study and other variables, but the researcher was unable to locate any studies that specifically focused on SC, EI, and PSS on ATSAP amongst senior secondary school students in Sokoto state of Nigeria. Moreover, the researcher was unable to locate

any research that investigated developing a model for the variables. In order to accomplish this, the purpose of the present study is to determine the link between the four variables and to construct a model of this relationship.

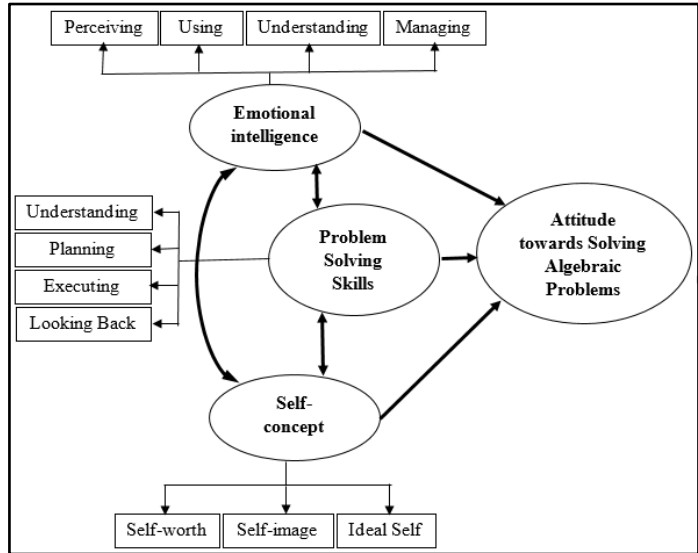

**Figure 1.** Conceptual framework of the research.

## 3. Research Methodology

The research design included a summary of all the activities contained in the research process from the objective formulation until the final analysis [73,74]. The researcher decided to adopt a quantitative research design that employs a survey method. Quantitative research design employs the study of a large population of respondents by selecting and studying a sample chosen from the population to ascertain the relative incidence, distribution, sociological, and psychological variables [75]. Often, survey research uses formal measures of behavior, including questionnaires, which are designed to be subjected to statistical analysis [76]. The use of a questionnaire was considered appropriate in this research in order to get to a large extent of sufficient responses from the respondents. The instrument used to measure the constructs in the present research was a self-developed questionnaire by the researcher. At first, a concept analysis was carried out, followed by an exploration of the literature in order to obtain data on the main constructs and dimensions of the variables under study. The study purpose is to examine the Sokoto state students' SC, EI, and PSS on their ATSAP.

The purpose of selecting a sample size is to obtain a group of subjects that will serve as a representative of the larger population. The objective of a study usually determines the method that is appropriate for the research. When a sample size is selected correctly, it becomes valid for the population and thus provides internal validity. According to Krejie and Morgan's Table (1970) for determining an appropriate sample size, the sample size of 377 was appropriate for the population of 21,839, as shown in Table 1.

**Table 1.** Distribution of students in the sample schools.

| S/No | School | No. of Students |
|---|---|---|
| 1 | School A in Danchadi District | 45 |
| 2 | School B in Gada District | 30 |
| 3 | School C in Kurawa District | 54 |
| 4 | School D in Gidan Igwe District | 66 |

**Table 1.** *Cont.*

| S/No | School | No. of Students |
|---|---|---|
| 5 | Sokoto E in Mabera District | 60 |
| 6 | Sokoto F in Ali-Akilu District | 65 |
| 7 | School G in Yabo District | 30 |
| 8 | School H in Sanyinna District | 27 |
| | **Total** | **377** |

## 4. Research Findings

As demonstrated in Table 2, the age of the respondents ranged from 11 to above 21 years old (three groups). The results indicated that 197 (52.3%) respondents belong to the age group of 11 to 14 years, 179 (47.5%) belong to the age group of 16 to 20 years, and one (3%) falls above 21 years of age. The majority of the students in secondary school two (SSII) were within the average age of 16 years and were believed to have the skills needed to read and understand written letters. The results of gender analysis revealed that 210 (55.7%) respondents were males and 167 (44.3%) of the respondents were females. There were more boys than girls in the secondary schools of Sokoto, Nigeria. This may be due to the fact that the girls were not encouraged to acquire Western education, but instead they were given out in marriage as compared to their male counterparts [77–79].

**Table 2.** Summary of demographic profile.

| Type | N | Factors | Frequency | Percentage (%) |
|---|---|---|---|---|
| Age | 377 | 11–15 years | 197 | 52.3 |
| | | 16–20 years | 179 | 47.5 |
| | | 21 years above | 1 | 0.30 |
| Gender | 377 | Males | 201 | 55.7 |
| | | Females | 167 | 44.3 |
| School Location | 377 | Urban | 269 | 71.4 |
| | | Rural | 108 | 28.6 |
| Parental Educational Level | 377 | Primary | 5 | 1.3 |
| | | Secondary | 71 | 18.8 |
| | | Diploma | 28 | 7.4 |
| | | Degree | 256 | 67.9 |
| | | Others | 17 | 4.5 |
| Parents' Occupation | 377 | Civil Servant | 276 | 73.2 |
| | | Farmer | 84 | 22.3 |
| | | Artisan | 8 | 2.1 |
| | | Others | 9 | 2.4 |

The school's location was classified according to whether it was in an urban or rural area. The analysis showed that 269 (71.4%) of the respondents were from urban schools, while 108 (28.6%) were from rural schools. The result of parents' educational level shows that 67.9% (N = 256) parents had acquired at least a degree, 7.4% (N = 28) had a diploma, 18.8% (N = 71) did not proceed with their education after the post primary level, 4.5% (N = 17) had not been to school, and 1.3% (N = 5) stopped their education at the primary (basic) level. From the results, it can be deduced that the parents of the majority of the students had acquired at least a degree as their academic qualification. It was believed that the parents of the respondents (being that they had acquired education to a certain level) knew the importance of supporting their children to succeed academically.

The distribution of the respondents based on their parents' occupation indicated that 276 (73.2%) were working parents, 84 (22.3%) were farmers, eight (2.1%) were artisans, and nine (2.4%) did not fall into the category of those aforementioned. Since more than 70% of

the parents of the respondents were working class, they were capable of paying their school fees and providing the school materials needed for their children to succeed in school.

*4.1. The Influence of Self-Concept (SC) on Attitude towards Solving Algebraic Problems (ATSAP) amongst Secondary School Students in Sokoto, Nigeria*

**Null Hypothesis 1:** *There is no significant influence between self-concept (SC) and attitude towards solving algebraic problems (ATSAP) among secondary school students in the state of Sokoto.*

The data analysis was performed by conducting a path analysis to determine the relationship between students' SC and their ATSAP by using SmartPLS3. This is presented in Figure 2.

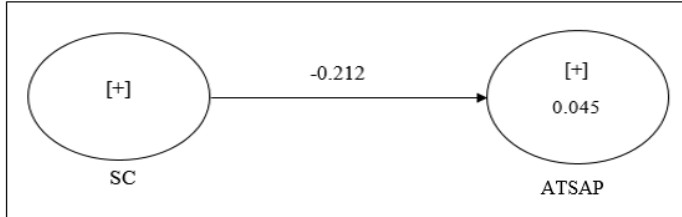

**Figure 2.** SC on ATSAP.

The path analysis results showed that SC had a significant direct negative influence on the students' ATSAP. As shown in Table 3, the relationship was negative with a path coefficient ($\beta$ = $-0.212$, t $\geq \pm 1.96$). According to [80], "The standard decision rule for a relationship between constructs to be significant is (*t*-value $\geq 1.96$ and *p*-value is $\leq 0.05$)". This was applied here to decide the significance of the path coefficient between the dependent variable and independent variable. Since the *t*-value was > $\pm 1.96$, the relationship between SC and ATSAP was significant and SC had influenced the students' ATSAP. The null hypothesis was rejected. In this study, the SC accounted for 0.045% of the variance in ATSAP.

**Table 3.** T-statistics and *p*-value of SC on ATSAP.

|  | Original Sample (O) | Sample Mean (M) | Standard Deviation (STDEV) | T Statistics (O/STDEV) | *p* Value |
|---|---|---|---|---|---|
| SC -> ATSAP | $-0.212$ | $-0.212$ | 0.046 | 4.565 | 0.000 *** |

*** The relationship is significant at $p < 0.001$.

*4.2. The Influence of Emotional Intelligence (EI) on Attitude towards Solving Algebraic Problems (ATSAP) amongst Secondary School Students in Sokoto, Nigeria*

**Null Hypothesis 2:** *There is no significant influence between emotional intelligence (EI) and attitude towards solving algebraic problems (ATSAP) among secondary school students in the state of Sokoto.*

The path analysis results on students' EI on their ATSAP showed that there was a relationship but not a significant one. The relationship was negative and not significant with a path coefficient ($\beta$ = $-0.043$, t < $\pm 1.96$) (Figure 3). Based on the t-statistics of 0.912 and a *p*-value of 0.362, it meant that EI had no influence on ATSAP. Only 0.002% of EI was explained in ATSAP. Therefore, the null hypothesis is accepted (Table 4).

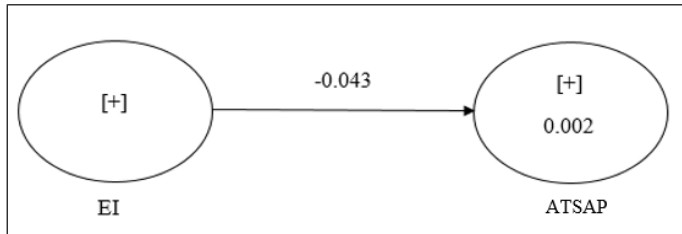

**Figure 3.** EI on ATSAP.

**Table 4.** T-statistics and *p*-value of EI on ATSAP.

|  | Original Sample (O) | Sample Mean (M) | Standard Deviation (STDEV) | T Statistics (|O/STDEV|) | *p* Values |
| --- | --- | --- | --- | --- | --- |
| EI -> ATSAP | −0.043 | −0.043 | 0.047 | 0.912 | 0.362 |

*4.3. The Influence of Problem-Solving Skills (PSS) on Attitude towards Solving Algebraic Problems (ATSAP) amongst Secondary School Students in Sokoto, Nigeria*

**Null Hypothesis 3:** *There is no significant influence between problem-solving skills (PSS) and attitude towards solving algebraic problems (ATSAP) among secondary school students in the state of Sokoto.*

Based on the path analysis in Figure 4, the result showed that the student's problem-solving skills had a direct positive relationship with their attitude towards solving algebraic problems. The relationship was significant and positive with a path coefficient (β = 0.105, t > ±1.96). This implied that the students' PSS influenced the attitude of the students towards solving algebraic problems (with a *t*-value of 2.002 and a *p*-value of 0.046). Since the *t*-value was >±1.96, it implied that PSS was significantly related to students' ATSAP (see Table 5). Therefore, the null hypothesis is rejected.

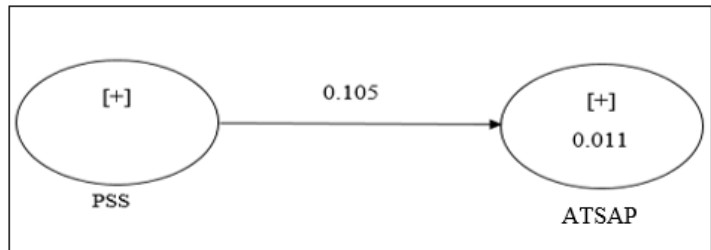

**Figure 4.** PSS on ATSAP.

**Table 5.** T-statistics and *p*-value of problem-solving skills on attitude.

|  | Original Sample (O) | Sample Mean (M) | Standard Deviation (STDEV) | T Statistics (|O/STDEV|) | *p* Values |
| --- | --- | --- | --- | --- | --- |
| PSS -> ATSAP | 0.105 | 0.104 | 0.053 | 2.002 | 0.046 ** |

** The relationship is significant at *p* < 0.01.

*4.4. The Relationship among Self-Concept (SC), Emotional Intelligence (EI) and Problem-Solving Skills (PSS)*

**Null Hypothesis 4:** *There is no significant relationship between self-concept (SC) and emotional intelligence (EI) among secondary school students in the state of Sokoto.*

The relationship between EI and SC was positively strong and significant with a t-statistics value of 9.083 which was >±1.96 (*p* value 0.000).

**Null Hypothesis 5:** *There is no significant relationship between emotional intelligence (EI) and problem-solving skills (PSS) among secondary school students in the state of Sokoto.*

EI had a relationship with PSS, but the relationship was not significant with a *t*-statistics value of 0.888, which was $< \pm 1.96$ (*p* value 0.375).

**Null Hypothesis 6:** *There is no significant relationship between self-concept (SC) and problem-solving skills (PSS) among secondary school students in the state of Sokoto.*

Students' PSS was related to their SC but the relationship was also not significant with a path coefficient of 0.014 and a *t*-statistics value of 0.296, which was $< \pm 1.96$ (*p* value 0.768).

Table 6 indicates that there exists a relationship among the three variables. Since SC and EI were significantly and positively related, an increase in SC would lead to an increase in EI and vice versa (see Figure 5).

**Table 6.** T-statistics and *p*-value of SC, EI, and PSS.

| | Original Sample (O) | Sample Mean (M) | Standard Deviation (STDEV) | T Statistics (|O/STDEV|) | *p* Values |
|---|---|---|---|---|---|
| EI -> PSS | 0.045 | 0.047 | 0.051 | 0.888 | 0.375 |
| EI -> SC | 0.416 | 0.419 | 0.046 | 9.083 | 0.000 *** |
| PSS -> SC | 0.014 | 0.017 | 0.046 | 0.296 | 0.768 |

*** The relationship is significant at *p* < 0.001.

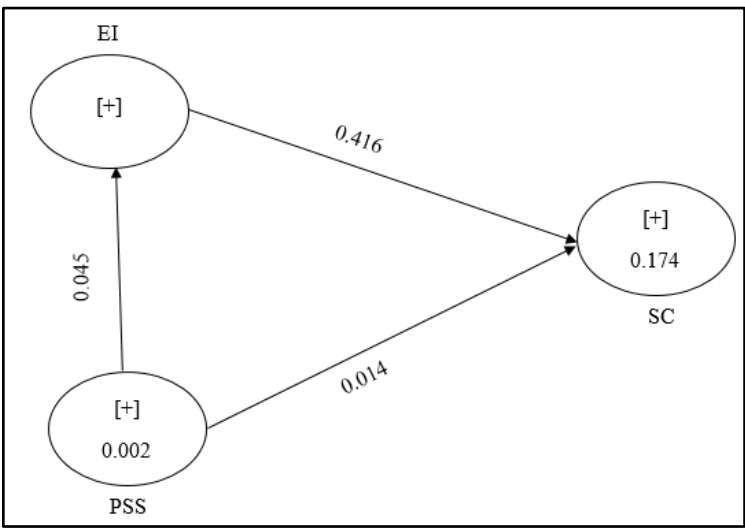

**Figure 5.** Structural model on the relationship of SC, EI and PSS.

To validate the proposed conceptual model, the path coefficient between two latent variables was assessed. Previous research has indicated that the path coefficient value needs to be at least 0.1 to account for a certain impact within the model [81,82]. Within the structural model, each path connecting two latent variables represented a hypothesis. Based on the analysis conducted on the conceptual model, it permits the researcher to accept or reject each hypothesis as well as understand the strength of the relationship between dependent and independent variables.

Assessment of the path coefficient in the current study, as shown in Figure 6, shows that some of the null hypotheses were supported while others were not supported. By using the SmartPLS3 algorithm output, the relationships between independent and dependent variables were examined. However, in SmartPLS3, in order to test the significant level, t-statistics for all paths were generated by using the SmartPLS3 bootstrapping function. Based on the t-statistics output, the significant level of each relationship was determined. Table 7 lists the path coefficients, observed *t*-statistics, and significance levels for all hypothesized

paths. Using the results from the path assessment, the acceptance or rejection of the proposed hypotheses was determined.

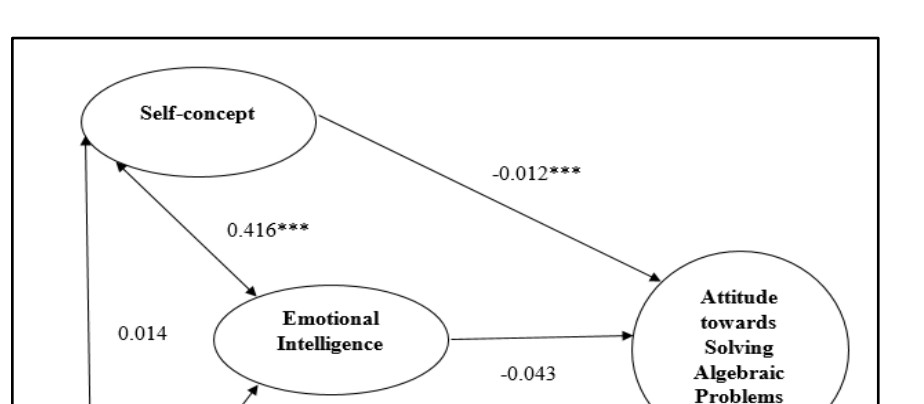

**Figure 6.** Structural model of the research. ** The relationship is significant at $p < 0.01$; *** The relationship is significant at $p < 0.001$.

**Table 7.** Path Coefficients, T- Statistics, and Significant Level.

| Relationship | Path Coefficient (β) | Observed T-Statistics | Significant Level (α) | Decision |
|---|---|---|---|---|
| **SC » ATSAP** | −0.212 | 4.565 | 0.000 *** | Rejected |
| **EI » ATSAP** | −0.043 | 0.912 | 0.362 | Accepted |
| **PSS » ATSAP** | 0.105 | 2.002 | 0.046 ** | Rejected |
| **EI » SC** | 0.416 | 9.083 | 0.000 *** | Rejected |
| **SC » PSS** | 0.014 | 0.296 | 0.768 | Accepted |
| **EI » PSS** | 0.045 | 0.888 | 0.375 | Accepted |

** The relationship is significant at $p < 0.01$; *** The relationship is significant at $p < 0.001$.

*4.5. Summary of Findings*

Based on the research findings, it was discovered that SC had an influence on the students' ATSAP. Students' ATSAP was found to be influenced by their PSS. Furthermore, it was demonstrated that EI did not influence the ATSAP. Moreover, PSS was found to have an influence on students' ATSAP. Finally, there exist relationships amongst SC, EI, and students' PSS. The relationship between SC and EI appeared to be very strong and positive, however, the relationship between SC and PSS was not significant, while the relationship between EI and PSS was also found to be non-significant. A structural model based on the responses of the students was developed. The model was based on four constructs, which were: SC, EI, PSS, and ATSAP. Table 8 below shows the summary of the findings based on the research hypotheses provided under each research question. Six hypotheses were tested where the study found that H01, H03, and H04 were not supported by the empirical findings, whereas H02, H05, and H06 were supported.

**Table 8.** Summary of Research Questions and Hypotheses.

| Research Questions and Hypotheses Statements | | Result |
|---|---|---|
| **Research Question 1:** Are there any significant influences of self-concept (SC), emotional intelligence (EI), and problem-solving skills (PSS) towards attitude towards solving algebraic problems (ATSAP)? | | |
| $H_{01}$ | There is no significant influence between self-concept (SC) and attitude towards solving algebraic problems (ATSAP) among secondary school students in the state of Sokoto. | Rejected |
| $H_{02}$ | There is no significant influence between emotional intelligence (EI) and attitude towards solving algebraic problems (ATSAP) among secondary school students in the state of Sokoto. | Accepted |
| $H_{03}$ | There is no significant influence between problem-solving skills (PSS) and attitude towards solving algebraic problems (ATSAP) among secondary school students in the state of Sokoto. | Rejected |
| **Research Question 2:** Are there any significant relationships between self-concept (SC), emotional intelligence (EI), and problem-solving skills (PSS)? | | |
| $H_{04}$ | There is no significant relationship between self-concept (SC) and emotional intelligence (EI) among secondary school students in the state of Sokoto. | Rejected |
| $H_{05}$ | There is no significant relationship between emotional intelligence (EI) and problem-solving skills (PSS) among secondary school students in the state of Sokoto. | Accepted |
| $H_{06}$ | There is no significant relationship between self-concept (SC) and problem-solving skills (PSS) among secondary school students in the state of Sokoto. | Accepted |

## 5. Discussions

The discussion of the findings in this section is written based on each research questions and the tested hypothesis.

*5.1. The Influence of Self-Concept (SC) on Attitude towards Solving Algebraic Problems (ATSAP) amongst Secondary School Students in Sokoto State*

In this study, SC was found to have a significant negative influence on students' ATSAP ($\beta = -0.212$, $t = 4.565$, $p < 0.001$). This meant that higher SC scores led to a decrease in ATSAP. This result was in agreement with previous studies' findings. For instance, ref. [83] examined attitude, SC, and achievement in basic science by using 360 junior secondary students in Ekiti state, Nigeria. There is a significant correlation between attitude, SC, and achievement in basic science. Moreover, ref. [84] found a significant correlation between skill learning attitudes and vocational SC among 270 Taiwanese junior high school students in the technical education program.

A study on academic SC and performance in mathematics by [52] showed that self-concept predicted academic performance in mathematics. Ref. [85] conducted a study amongst 350 senior secondary students selected through a stratified proportionate random sampling and also discovered that mathematics SC significantly predicted students' attitude towards mathematics in Obio-Akpor Local Government Area of Rivers State. Most of the previous research was in agreement with the current study, with a significant influence of SC on ATSAP. It is worth noting that in the previous studies conducted on SC and students' attitudes, there was a significant positive influence. On the contrary, in the present study, the influence of SC on students' attitude towards solving algebraic problems was negatively significant. This simply implied that an increase in SC would lead to a decrease in attitude towards solving algebraic problems and vice versa. The results revealed that the respondents had a strong belief about what they "can and cannot do" and instead had a negative ATSAP.

The result of the study conducted by [83] indicated that most of the students had a positive attitude towards learning basic science. As such, their SC had a significant

and positive influence on their attitude towards basic science. The negative influence of students' SC on ATSAP in the current study was the result of the students' negative attitude demonstrated towards solving problems. Students' positive attitude towards solving algebraic problems in the current study, SC would have a positively significant influence on the students' ATSAP. The descriptive analysis results showed that the students' negative attitude towards solving algebraic problems had led them to avoid anything related to algebraic problems. Most of the students are lacking in self-confidence to solve the presented algebraic problems.

Most of the students gave up on given tasks due to their negative attitude towards solving algebraic problems. According to [86], students' problem-solving beliefs were very strong and fundamental in determining their achievement in solving problems. Whenever their problem-solving belief increases, their achievement in solving problems also increases. Their findings were in agreement with the current study results. In the current study, the majority of the respondents had a strong belief that solving algebraic problems was very difficult, confusing, and required them to think deeply. This has brought about the students having a negative attitude towards solving algebraic problems The students need to be supported, encouraged, and challenged in order to develop a positive SC towards solving algebraic problems.

### 5.2. The Influence of Emotional Intelligence (EI) on Attitude towards Solving Algebraic Problems (ATSAP) amongst Secondary School Students in Sokoto State

The findings revealed that EI had a negative but non-significant influence on students' ATSAP ($\beta = -0.043$, $t = 0.912$, $p > 0.05$). This result was in agreement with the study of [87], which also found that attitudes towards computer-based instructions correlated negatively to students' EI. Moreover, ref. [88] conducted a study on the relationship between EI and students' attitude towards computers. The study was conducted at a polytechnic utilizing engineering students. The results unveiled that the relationship between EI and attitude towards computers was positive but weak and not significant. Meanwhile, the relationship between EI and attitude towards solving algebraic problems in the current study was weak, not significant, and was also negative. The current study showed that the students had low EI with a negative ATSAP. This implied that an increase in their EI would lead to a decrease in their display of negative ATSAP. Work conducted by [89] also supported these results. It was shown that EI was negatively correlated with pain knowledge and attitudes, and the correlation was not significant.

Similarly, ref. [90] revealed that students with a low level of EI life had a negative correlation with their attitudes towards mathematics learning. In contrast to the current study, there were some previous studies conducted on students' attitude and EI [91–93]. For instance, in a study conducted by [91] on students' EI and their attitudes towards science in equity education. The results revealed that there was a positive and significant relationship between students' attitude towards science and EI. Supporting the result of [91] was the study by [87], where the study investigated EI and hospitality students' attitudes towards e-learning. The results indicated that there existed a significant positive correlation between the two variables; as students' total EI increased, their overall attitude towards e-learning also increased.

In comparison to the results of the previously conducted study on students' EI and their attitudes, it has been uncovered that the students in the current study had low EI and a negative attitude towards solving algebraic problems. Therefore, the negative influence of the EI on their attitude towards solving algebraic problems. The influence of EI on ATSAP was not significant, which was possibly due to the students' low EI.

### 5.3. The Influence of Problem-Solving Skills (PSS) on Attitude towards Solving Algebraic Problems (ATSAP) amongst Secondary School Students in Sokoto State

The present study indicated that the students' PSS had a positive influence on their ATSAP ($\beta = 0.105$, $t = 2.002$, $p < 0.05$). This finding was aligned with the previous studies that examined the relationship between students' attitudes towards mathematics and their

problem-solving skills. The results showed that there was a strong positive correlation between students' attitudes towards mathematics and their problem-solving skills. The findings are also in line with the work of [15], where the study investigated attitudes and problem-solving skills in algebra amongst Malaysian matriculation college students. The outcome of their findings indicated that there was a moderate relationship between the students' attitudes and problem-solving skills in algebra. Moreover, they recorded low attitudes for some students, which they believed may be due to those students' unwillingness to solve problems and also the lack of perseverance to solve the problems.

It is important to note that though the present study showed that students' ATSAP was influenced by their PSS, the students' attitude based on the descriptive analysis showed a negative attitude towards solving algebraic problems. This may be due to their low problem-solving skills, which prompted their unwillingness to solve problems by themselves. They reported that students had difficulty solving problems and also had difficulty with algebraic concepts. For instance, most of the students failed to understand the problems and had not mastered the basic skills for solving problems [94,95].

Much of the previous research also supported the influence of problem-solving skills on attitude and found significant relationships between them [71,96–101]. For instance, [101] looked at social problem-solving skills and dysfunctional attitudes with a risk of drug abuse amongst dormitory students at Isfahan University of Medical Science. In the study, it was found that social problem-solving skills had a significant influence on dysfunctional attitudes. Moreover, a study looking at the relationship between problem-solving skills and academic achievement, carried out by [99] that utilized primary school pupils, discovered that the pupils' PSS skills were significantly related to their academic achievement. The study showed that the students who had relatively higher school success also had a relatively high level of PSS. In addition, a study on problem solving ability in relation to academic achievement [102] unveiled that problem solving ability is significantly and positively correlated with students' academic achievement. This result invariably meant that a high level of ability to solve problems led to a high level of achievement academically and otherwise. The author of [103] also conducted a study on the problem-solving abilities of secondary school students in relation to their attitudes towards mathematics. His study's results supported the result in the current study; there is a relationship between the problem-solving ability of secondary school students and their attitudes towards mathematics.

The current study's descriptive analysis showed that the Sokoto state students had a negative attitude towards solving algebraic problems and also scored low in their problem-solving skills test. Their problem-solving skills positively predicted their attitude towards solving algebraic problems. However, ref. [104] further affirmed this in their study on problem-solving ability and the academic achievement of high school students. Utilizing 250 students studying in their 10th class of high school affiliated to the CBSE of Rohtak district by employing a random sampling. The problem-solving ability test (PSAT), developed by L. N. Dubey, was used as the instrument in the study. The findings of the study revealed that problem-solving ability had a significant effect on the academic achievement of high school students. Similarly, ref. [21] investigated the effect of attitude toward problem solving in mathematics achievement at the Malaysia Institute of Information and Technology (MIIT) University KL, which included 153 Diploma and bachelor's degree students, and the survey focused on students' patience, willingness, and confidence. The findings revealed that students' levels of patience, confidence, and willingness to solve problems were moderate, and that there was a significant relationship between attitude in problem solving and mathematics achievement.

### 5.4. Relationship among Self-Concept, Emotional Intelligence and Problem-Solving Skills

The second research question focused on eliciting empirical evidence on the relationship that exists between SC, EI, and PSS. The purpose of investigating the relationship among the three independent variables was to empirically demonstrate how they relate to

one another. The researcher was interested in trying to find out if an increase in one of the variables would lead to an increase in another and vice versa, or if what happened to one variable would not affect the other variable. The relationship analysis findings exposed that a strong and positive relationship exists between SC and students' EI ($\beta = 0.416$, $t = 9.083$, $p < 0.05$). This was in line with the earlier research of [59], who looked at the connection between students' EI and a positive SC. Their study revealed that a positive and significant relationship exists between EI and students' SC. Similar results were in agreement with the previous studies [34,105–108]. Ref. [105] examined the link between EI and SC and love failure. The study employed 84 participants, comprising those who had experienced failure in love and those who had not. The study results indicated that there was a relationship between EI and SC because individuals with an experience of failure in love possessed a low degree of EI. Those individuals with experience of failure in love possessed a lower SC. This implied that an increase in EI had a corresponding increase in SC in relation to experience of failure in love.

The current study demonstrates that an increase in EI led to an increase in SC in relation to the students' ATSAP. Ref. [34] also considered the influence of EI on SC by using 134 students from the University of Almeria. The outcome uncovered that EI significantly and positively influenced SC. The findings confirmed the current study findings, which further confirmed the findings of the works of [109] that found a relationship between EI and SC. Furthermore, ref. [110] studied the relationship between EI and SC of 150 B.Ed students, where their findings showed that there was a significant relationship between EI and SC of the B.Ed students.

The current study results were consistent with the study of [111], who found that the correlation between the SC and EI reflected a moderate positive correlation, which signified the tendency for an increase in SC score with the increase of the score of EI and vice versa. The study involved 150 adolescents and found that SC was the foundation of personal growth and helped in developing EI at the crucial stage of adolescence. Likewise, ref. [112] discovered a significant and positive relationship between SC, EI, and the academic achievement of students in their study on the relationship between EI and SC with students' academic achievement by using 99 students from Qaen nursing and midwifery school.

However, the current study's results disclosed that EI and PSS were not significantly related ($\beta = 0.045$, $t = 0.888$, $p > 0.05$). This outcome was in agreement with the previous research [66], where it was discovered that there was no significant correlation between other components of EI and problem solving. This was in contrast to the current study findings. Ref. [113] found that problem solving and emotional education in initial primary teachers were significantly related. The study conducted by [17] also unveiled that there exists a positive and significant relationship between problem-solving ability and academic achievement of secondary school students in the Maldives. It can be agreed that the higher the problem-solving ability of the students, the higher their academic achievement. Research conducted by [114] regarding cognitive and meta-cognition (2015) on EI and problem-solving achievement of chemistry students. The results of the findings indicated that EI was significantly related to the students' achievement in chemistry problem solving. Furthermore, ref. [115] research findings also contradicted with the current study's findings. They investigated problem solving and EI on the decrease of the third-grade girl students' aggression at the Rajaee guidance school in Tehran. The results indicated that problem solving and EI were significantly related to decreasing aggression in third grade girl students in Tehran. Other research that supported the results was the works of [116–120]. A student who could recognize and control their emotions could exhibit more positive approaches to problems and, accordingly, could solve them more easily [117].

Furthermore, the relationship between SC and PSS was investigated. The outcome indicated that the relationship between them was not significant ($\beta = 0.014$, $t = 0.296$, $p > 0.05$). This finding aligned with the findings of [121] that found low-achievement students' SC was not significantly related to their problem-solving ability. In addition, ref. [122] investigated SC and academic achievement in mathematics, and the results

revealed that SC did not have a significant relationship with academic achievement in mathematics. Likewise, ref. [57] found that students' SC did not relate to their academic achievement in mathematics. Similarly, ref. [56] carried out research on the influence of students' SC on their academic performance in Elmina Township. The results showed that SC did not directly predict students' academic performance. Although, the findings of the current study contradict the findings of [51,69]. Both of them, in their separate research, discovered that both SC and students' problem-solving ability were positively and significantly related to each other.

### 5.5. The Proposed Model

Figure 7 shows the proposed model, the study findings implication of SC, EI, and PSS all in relation to ATSAP amongst secondary school students in Sokoto state, Nigeria. This model was developed based on the responses of the students. In this research, it was discovered that students' SC and EI were the most significant variables that could influence students' ATSAP among secondary school students in Sokoto state. The research outcome uncovered that about 70% of the students had a good and positive SC of themselves and about 63% had a positive EI, although still at a low level. This implied that since there was a positive relationship between SC and EI, therefore, an increase in students' SC would lead to a corresponding increase in their EI. However, the inverse relationship between SC and ATSAP showed that an increase in students' SC would lead to a decrease in their ATSAP, and it was discovered in the study that Sokoto state secondary school students had a negative ATSAP. As such, an increase in their SC and EI would lead to their attitude towards a positive direction. Furthermore, the findings showed that there was a positive relationship between students' ATSAP and their PSS. This therefore meant that any positive change or increase in attitude would lead to an increase in their problem-solving skills.

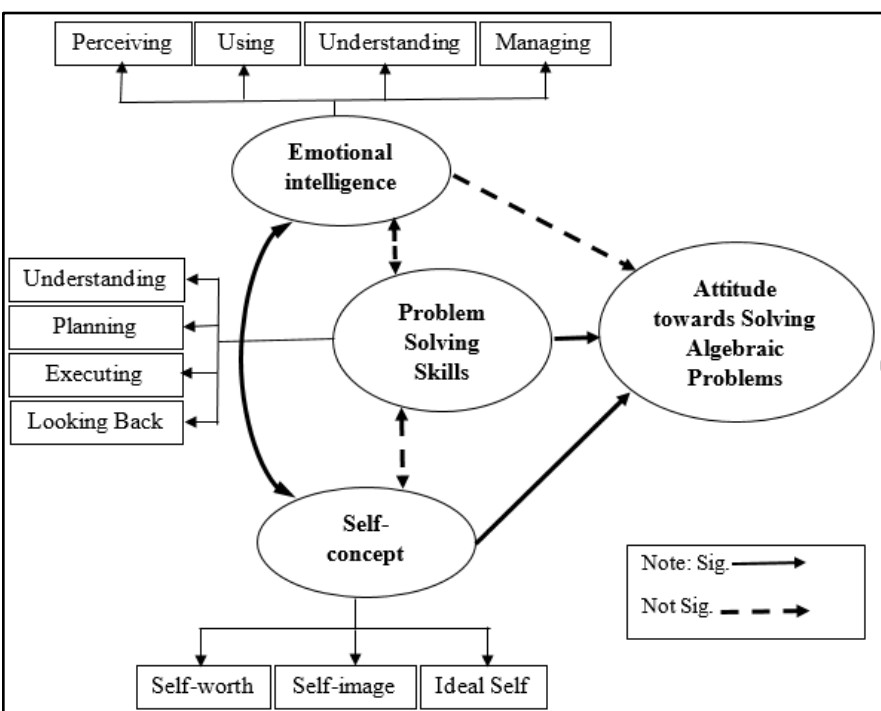

**Figure 7.** The Proposed Model.

Overall, the model suggests that there is a relationship that exists between all the variables under study, giving important and meaningful consideration to students' SC and its influence on students' ATSAP. Moreover, based on the strong positive relationship that exists between EI and SC in the present study, it implied that once there was an increase in student SC, it would automatically lead to an increase in their EI. Meanwhile, the model

also showed that the student's ATSAP was negative. According to [86], environmental settings could affect the way a person views a particular object, positively or negatively. This study, therefore, based on the outcome of the model, suggests that parents and teachers should assist these children to have a more positive SC of themselves so as to improve their attitude.

The proposed model would stand as a mirror and a pointer that depicted how students' SC, EI, and PSS could be harmonized to produce an enhanced and positive mathematics learning classroom atmosphere. This would enable the students to reach their full potential in solving algebraic problems and developing self-confidence. However, it would also lead to the scientific and technological development of Nigeria.

## 6. Limitation of the Study

In the process of conducting the present research, some limitations were encountered, which may have affected the outcome of the findings. Among them is the fact that the study was conducted among public secondary school students in Sokoto state, where private schools were exempted, the findings of the study may not be generalized to the entire population of secondary school students in Nigeria. Secondly, the study was restricted to only secondary school students. The study further concentrated only on the algebraic aspect of mathematics and not mathematics in general. The researcher's bias could be a limitation for the study because much time has been spent on the variables under study. The study only focuses on some aspects of self, used only the factors in ability models to look into students' emotional intelligence and concentrated on Gestalt problem solving to look into the student's problem-solving ability, where as there are other models which may be, if used could have yield better results. The model provided in the study does not suggest how the self, emotions, and confidence of students can be enhanced, but only limits itself to showing how the variables are related and can enhance performance in life if well taken care of.

## 7. Suggestions for Further Study

Based on the findings of the study, the following suggestions were put forward. There is a need to conduct this research on a larger sample in Nigeria by extending the study to other zones and other subjects, and even on different levels of study in terms of comparison. It is suggested that another study that will present the modalities of how to successfully implement the application of these variables into the classroom teaching and learning process should be adopted. A similar study could be carried out in other countries so as to subsequently determine the similarities and differences in the model developed in the present study and the ones that would be developed in another country. Other factors than the ones studied in the present research that could also enhance students' problem-solving ability and help inculcate a positive attitude towards solving problems should also be carried out. There is a need to develop a curriculum that will take all the variables under study into the activity of teaching and learning in the classroom into consideration.

## 8. Conclusions

In conclusion, this study has been able to provide a model for enhancing a positive attitude in students towards solving algebraic problems. The validated model was produced based on the students' responses and structured under four constructs; whereby major findings of the study include but are not limited to the following, which are (1) the study has assisted in developing a model that showed how related the variables under study were and how they could enhance students' PSS. Enhancing PSS could be achieved once the ATSAP is positive; (2) a student's SC significantly affected the display of attitude that a student will have towards things, even though in the present study one of the dimensions of the students' SC was low, yet it significantly relates to their ATSAP. This means that the way the students view things, how they aspire to achieve, and how confident they are in achieving those things determine, the attitude they will have towards things; (3) two

dimensions of their EI were high, while the other two were at low and moderate levels, yet their EI was related to their ATSAP. Although the relationship between EI and ATSAP was not significant; and (4) the study has helped to develop a hypothesized model of the relationship amongst SC, EI, and PSS with relation to ATSAP. Therefore, the study found that the four constructs could be combined together to enhance the students' performance in algebra.

**Author Contributions:** Conceptualization, A.H.A. and E.J.; methodology, N.S.; validation, M.A., C.A.T. and Z.M.A.; formal analysis, U.H.A.K.; writing—review and editing, S.N.S.A.R. All authors have read and agreed to the published version of the manuscript.

**Funding:** This work was supported by the UTM Fundamental Research Grant Q.J130000.3813.21H85, and in part by the Ministry of Higher Education Malaysia and Universiti Teknologi Malaysia.

**Institutional Review Board Statement:** Not applicable.

**Informed Consent Statement:** Informed consent was obtained from all participants involved in this study.

**Data Availability Statement:** Not applicable.

**Conflicts of Interest:** The authors declare no conflict of interest.

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
