# Peer review of "Relationship between Self-Concept, Emotional Intelligence and Problem-Solving Skills on Secondary School Students’ Attitude towards Solving Algebraic Problems"

_sustainability, doi:10.3390/su142114402_

Round 1

Reviewer 1 Report

In title: Delete the word "the" as well as initials like SC, EI, PSS and ATSAP. Include information about the main characteristic of participants.

Abstract: Consider if including values is really necessary or not.

Keywords: Avoid repeating words from those which already are in title.

Introduction: Review how quotes have been included in text. In its current way, it makes difficult to read the paper. Justify lines 67-69 with a quote. The theoretical framework must be updated, using references from 2022, 2021 or 2020. Reflect if it is adequate to split introduction in only one subsection. Including other subsections or connecting the current content with the whole introduction are potential solutions.

Conclusion: Provide more information about the limitations of the study as well as potential future works that could be developed from these paper findings.

Author Response

Dear reviewer,

We have amended the manuscript according to the comments.

Thank you

Reviewer 2 Report

·        This paper explored factors impacting an important subject for students. The 25 pages can be shortened a lot. There are a lot of unnecessary details. Descriptions are often redundant. The flow is hard to follow. Transitional words like “however” are often used inappropriately when there were no contrasts of ideas. Also all citations only have numbers when oftentimes author names should be mentioned.

·        Abstract did not start with a brief background description.

·        Line 14 aimed to, line 17 aimed at.  Consider using a different phrase for line 17 such as “intended to develop”

·        Line 17, consider deleting “and utilized”

·        Consider deleting line 33-35 “This….nation building” as in line 39 you talked about the importance of math again from a different angle.

·        Lines 46-51 the information presents the same message as lines 44-45. Consider deleting redundant info.

·        Line 57, keep it simple: “Therefore, the ultimate aim…”

·        Lines 62-69, repetition of the same info.

·        Line 70, why using “however”?

·        Lines 78-79, self-concept, self-efficacy…and attitudes, 4 factors are mentioned here. Why does the next paragraph start with attitude? What is the flow here?

·        Lines 80-86, consider deleting all

·        Line 91 and line 93, both lines have the word “therefore”.  It is not necessary

·        Line 92, not unconnected,  how about just “connected”?

·        Lines 99-100, why stating it again?

·        Line 102, why “therefore”? What is the logic of the flow?

·        Line 107, why just now talking about self-concept? What order are you following?

·        Line 107-136, very lengthy, consider making it concise and straight to the point

·        Line 126, why using “however”, what contrasts are there between the sentence before and after?

·        Line 133, missing period after [29].

·        Line 134, while on the contrary.  Delete “while”

·        Line 137, it starts with “it is therefore” and “no doubt” like line 57. Why using “therefore” so often?

·        Line 146, delete “on”

·        Line 148 assist thoughts or assist thinking?

·        Lines 153-154, need a citation

·        Lines 159-163 not understanding the flow here

·        Line 164, who are “their”?

·        Line 197, why “however”?

·        Line 200, I see“on the other hand”, where is “on one hand” then?

·        Lines 210-211, two sentences can be shortened to one: “On the contrary, studies by xxx [55]  and [56] found that SC…performance.”

·        Line 218: in detail not in detailed.

·        Line 215 mentioned 3 variables. Line 218 mentioned only 2 variables. How do you make it flow smoothly?

·        Conceptual framework section starting from line 222: 4 constructs and 11 sub-constructs were mentioned but there was no description of subconstructs at all. Why including them?

·        Line 230, it says “based on this research”? What research, the previous research or your own research? Your framework should not be based on your own current study.

·        Research methodology, from line 250 to 274, this section should be greatly simplified. There is no need to distinguish between three basic approaches at the beginning. There is no need to explain in detail what a survey method is. Those are so commonly used in research. Details are unnecessary.

·        Respondent selection, line 276 to line 281. Consider deleting them. Go straight to the point and explain how they were selected.

·        Table 1, how did you determine the numbers for each district? Here we need details.

·        3.2 I would rename the subheading to “construct reliability”. Avoid “findings” until you get to the real Results section.

·        Table 2-5 are unnecessary. They can be just explained in text. The info. can be provided in appendices instead. Also which construct is table 3? The name does not describe the construct as it does in Table 2.

·        Line 295, since no subconstructs were explained with Figure 1, readers now find self-worth, self-image, and ideal-self and wonder what they are for.

·        Lines 306-307, why repeating the same info. every time? Only needs to mention it once. Same for lines 315-316.

·        Line 319, why a separate subheading for PSS?

·        Delete line 338.

·        Line 342, “as such” are used often in the text. Delete

·        Line 343, “however” again

·        Line 349, why “consequently”, consequence from what?

·        Line 355, delete “were the number that”

·        Line 363, “however” again. How about just “Since more than 70% of …”

·        Line 372: how about “a part analysis was performed to determine…” so that we don’t use “by doing” twice in one sentence.

·        Line 381-385, no need to explain in such detail how to reject a hypothesis.

·        Line 386, why “was not rejected”? A significant p value does not indicate we reject the null hypothesis? Please explain.

·        Line 394. why not accepted?

·        Line 408,  why not rejected?

·        Line 437, grammatically wrong sentence

·        Table 11, not understanding the decisions here.

·        Lines 450-452, consider deleting

·        Line 454, consider saying “selected as participants” not “used as participants”

·        Line 458, there exist

·        Line 461, “however”

·        Line 469, the heading should be “Discussion” not “Main findings”

·        Combine line 480-482. Authors [83] conducted a study… and found a significant…

·        Restructure line 484 -487 in the same manner. What they conducted should be mentioned first before you mention what they found.

·        Line 485 “submitted” was used several times in Discussion. Consider changing it to “discovered”, “found”, “claimed” or some other words

·        Line 488, delete “conducted results”.

·        Line 501, use of “however” again

·        Line 505, “based on the interview”, what interview?

·        Line 514, “however”

·        Line 519, “their submission”, you mean “their finding”

·        Line 534, how about “weak, insignificant and negative”?

·        Line 536, delete “though”

·        Line 537, delete “instead”

·        Line 539, delete “that also”

·        Line 547, the sentence was not completed yet: education, the results…

·        Line 557, “which believed due to…” how about just “possibly due to…”

·        Line 558, consider deleting the sentence

·        Line 571, lack of

·        Line 574, negative what?

·        Line 576, delete “this was supported by the.”

·        Line 580, delete “on”

·        Line 581, what “attitude” do you refer to?

·        Line 583, Isfahan University of Medical Science

·        Line 600, “however”

·        Line 609, “whereby” is used a lot. See line 622, 626, 639 etc.

·        Line 611, moderate. It also showed…

·        Line 633, “however”

·        Line 641, “submitted” – see comment 61

·        Line 650, 652, which one is in agreement and which is in contract to?

·        Line 659, citation format

·        Line 665, “the latter” refers to?

·        Line 671, delete “the result in”

·        Line 679 “both of them” not “both them”

·        Line 702, incomplete sentence

·        Line 710, who is Rogers?

·        Missing limitations of the study and future implications in the Discussion section

·        The bullet points in the conclusion does not look like a normal journal format.

Author Response

(The authors gave the same response as above.)

Reviewer 3 Report

I think that theoretical basis of the research should be improved:

1. The concept of EI. It is risky to research among 11-14 year students, because in such age empathy and general EI start to develop.

2. It is risky to research connection of algebraic skills and EI. No theoretical reason. The inconsistency in the research results, presented by the authors, should be deeper analysed. EI is related to the social skills, maths and science subjects to analytical skills and logic.  Try to find other explanation.

3. I suggest to enrich references (and the theoretical background as well):

-M. Spitzer, Learning: The Human Brain and the School of Life

-Knysh, A. ., Romanovskyi, O., Pidbutska, N., & Shtuchenko, I. (2020). Connection Between Personal Perfectionism and Efficency of Students’ Learning Activities. Journal of Education Culture and Society, 11(1), 136–145. https://doi.org/10.15503/jecs2020.1.136.145

-Shibuya, E. (2020). Emotional factors in senior L2 acquisition: A case study of Japanese speakers learning Spanish. Journal of Education Culture and Society, 11(1), 353–369. https://doi.org/10.15503/jecs2020.1.353.369

-Kyrylenko, T. ., & Shamlyan, K. . (2020). Distinctive features and types of interrelation between the indices of emotional and volitional personality sphere. Journal of Education Culture and Society, 5(2), 102–111. https://doi.org/10.15503/jecs20142.102.111

Author Response

(The authors gave the same response as above.)

Round 2

Reviewer 1 Report

Dear authors,

I appreciate your effort and, in my humble view, this new version is better than the previous one. I only have two comments:

1.- I wonder if quotation in text guidelines are being fitted or not.

2.- Review the numbers of section (6.0 and 7.0 instead of 6 and 7).

Author Response

Dear reviewer

We have done our best to amend the manuscript based on your comments.

Thanks and we truly appreciate that.

Reviewer 2 Report

Thank you for making the revisions. The paper does flow better than the previous version. Not all suggestions were addressed though. Here are some further suggestions: 

1. Intro: Line 45-46 is in a separate paragraph. consider combining it with the next. Line 101, is it "not connected" or "connected"?

2. The methods section never mentioned an interview but then Discussion line 541 and line 690 discussed the interview. What is the interview? Why was it conducted? This was not described in Methods. 

3. line 246-250, not necessary. consider deleting

4. Line 260 "the objective of ...validity", not necessary. consider deleting

5. Table 1, it is still not discussed in the text how the no. of students for each district was determined. Did you use proportional sampling based on the student population in each district?

6. Table 2-5, as mentioned, can be in appendices instead of the body of the paper.  How many items for each construct should be clearly described. Some descriptions say "there are x items", some don't.  Also each table should be followed with caption such as Table 2 SC construct and its sub-construct. What is currently in there is the overall result which should be at the bottom as a "Note". 

7. line 327, restructure the sentence to eliminate two "by..."

8. line 337 explains the decision rule but line 341 says "the hypothesis was not rejected." T is 4.565 and p is 0.000. Why not rejecting the null hypothesis? In stats, as far as I know, when we say we reject or fail to reject, we are referring to the null hypothesis.  Line 349, line 363... and table 11 Decision column, it all seems to be the opposite. 

9. Table 12 - 2 research questions are listed here. These questions should be clearly stated in Introduction too in the exact same manner. 

10. Line 435, delete the extra "They."  Original comments for 5.2-5.4 were not addressed.  The word "whereby" is used repeatedly in Discussion. Consider changing it up. 

11. I personally do not like the bullet format for a conclusion. 

Author Response

(The authors gave the same response as above.)

Round 3

Reviewer 2 Report

Research question 1: Does the factors ...have influence - grammatically incorrect sentence.  Also for research questions, it is a common practice to avoid yes/no questions, instead ask "what", "how", "why".

The authors added 3 lines about the interview (281-283). However, this does not tell me how many questions are asked, what questions are asked, how many participated in the interview, how were the participants selected, how were interview data analyzed. The findings from the interview were not presented in the results section but it is discussed in Discussion. 

If you have a quantitative survey and a qualitative interview, it is a mixed-methods study. For mixed-methods studies, there should be an explanation of why use both quantitative and qualitative approaches, whether data are collected sequentially or concurrently, which set of data is given priority in analysis, and an integration of data in reporting. Please look at Designing and Conducting Mixed Methods Research by Creswell and O'Cathain's Good Reporting of a Mixed Methods Study. If this is too much to change, take out the interview information all together. 

I still don't see the explanation on the "Decision" column why p of 0.000 renders an "accepted" decision. When you accept or reject a hypothesis, it is accepting or rejecting the null hypothesis. I still see all the Cronbach's alpha tables in the text. I personally don't see the value of that many tables on Cronbach in the text but if the editors are okay with them, that is fine with me.   

Author Response

Dear reviewer

We have tried our best to amend the manuscript based on your constructive comments. Thanks and best regards,

Halim
